# Cathepsin D—Managing the Delicate Balance

**DOI:** 10.3390/pharmaceutics13060837

**Published:** 2021-06-05

**Authors:** Olja Mijanovic, Anastasiia I. Petushkova, Ana Brankovic, Boris Turk, Anna B. Solovieva, Angelina I. Nikitkina, Sergey Bolevich, Peter S. Timashev, Alessandro Parodi, Andrey A. Zamyatnin

**Affiliations:** 1Institute for Regenerative Medicine, Sechenov First Moscow State Medical University, 119991 Moscow, Russia; olja.mijanovic@gmail.com (O.M.); boris.turk@ijs.si (B.T.); nikitina_a_i@staff.sechenov.ru (A.I.N.); timashev_p_s@staff.sechenov.ru (P.S.T.); 2Institute of Molecular Medicine, Sechenov First Moscow State Medical University, 119991 Moscow, Russia; asyapeti@gmail.com; 3Laboratory of Nanomedicine, Department of Biotechnology, Sirius University of Science and Technology, 1 Olympic Ave, 354340 Sochi, Russia; 4Department of Forensic Engineering, University of Criminal Investigation and Police Studies, 11080 Belgrade, Serbia; ana.brankovic@kpu.edu.rs; 5Department of Biochemistry and Molecular and Structural Biology, J. Stefan Institute, 1000 Ljubljana, Slovenia; 6Faculty of Chemistry and Chemical Technology, University of Ljubljana, 1000 Ljubljana, Slovenia; 7Semenov Institute of Chemical Physics, Russian Academy of Sciences, 119991 Moscow, Russia; ann.solovieva@gmail.com; 8Department of Human Pathology, Sechenov University, 119991 Moscow, Russia; bolevich2011@yandex.ru; 9Faculty of Chemistry, Lomonosov Moscow State University, 119234 Moscow, Russia; 10World-Class Research Center “Digital Biodesign and Personalized Healthcare”, Sechenov University, 119991 Moscow, Russia; 11Belozersky Institute of Physico-Chemical Biology, Lomonosov Moscow State University, 119992 Moscow, Russia

**Keywords:** cathepsin D, lysosome, regulated cell death, neurodegenerative disease, malignant tumor, diabetes, inhibitors

## Abstract

Lysosomal proteases play a crucial role in maintaining cell homeostasis. Human cathepsin D manages protein turnover degrading misfolded and aggregated proteins and favors apoptosis in the case of proteostasis disruption. However, when cathepsin D regulation is affected, it can contribute to numerous disorders. The down-regulation of human cathepsin D is associated with neurodegenerative disorders, such as neuronal ceroid lipofuscinosis. On the other hand, its excessive levels outside lysosomes and the cell membrane lead to tumor growth, migration, invasion and angiogenesis. Therefore, targeting cathepsin D could provide significant diagnostic benefits and new avenues of therapy. Herein, we provide a brief overview of cathepsin D structure, regulation, function, and its role in the progression of many diseases and the therapeutic potentialities of natural and synthetic inhibitors and activators of this protease.

## 1. Introduction

Lysosomes are spherical acidic vesicles found in all mammalian cells but erythrocytes. Their primary role is the disposal and the recycling of exhausted and deteriorated macromolecules and organelles, along with the digestion of alien structures supplied by endo- and phagocytosis [1]. Lysosomes contain more than 60 hydrolytic enzymes, including proteases, lipases, nucleases, glycosidases, phospholipases, phosphatases, and sulfatases [2]. Their acidic environment (pH 4–5), optimal for the activity of these enzymes, is maintained by a vacuolar-type H^+^-adenosine triphosphatase (ATPase) pumping protons from the cytosol into the lysosome lumen [3]. Cathepsins are the main mammalian lysosomal proteases, and they are classified into three groups according to their catalytic mechanism: Cysteine cathepsins (B, C, F, H, K, L, O, S, V, W, and X), serine cathepsins (A and G), and aspartic cathepsins (D and E) [4]. Most of them possess endopeptidase activity (cleaving internal peptide bonds), whereas some possess exopeptidase activity (cleaving off amino acid residues at the N- or C-terminal domains). Some cathepsins are both endopeptidases and exopeptidases, and their activity depends on their localization and environmental pH. There are many levels of cathepsin activity regulation, including biosynthetic processes, trafficking to lysosomes and other compartments, (auto-)proteolytic activation cleavage, and endogenous inhibitors [5,6,7,8]. They take part in tissue and bone remodeling, major histocompatibility complex class II-mediated antigen processing and presentation, protein, hormone, and neuropeptide processing, wound healing, apoptosis, as well as in disease development and progression, including cancer, inflammation, atherosclerosis, rheumatoid arthritis and neurodegeneration [9,10,11,12,13].

The aspartic protease cathepsin D is the most abundant lysosomal protease [14]. The human gene (*CTSD*) is located at the 11p15.5 region and contains 9 exons [15]. A mixed promoter controls cathepsin D gene expression, enabling both TATA-independent (binding site for specificity protein (Sp) 1) and TATA-dependent transcription initiations. Also, it was reported that its transcription might be activated by estrogen [16]. The encoded human cathepsin D protein comprises 412 amino acid residues. Following the removal of the signal peptide and two Asn glycosylations (134 and 263pre-pro-cathepsin D numbering) in the endoplasmic reticulum, the propeptide is transported to the endolysosomal compartment via mannose-6-phosphate (M6P)-dependent or independent (via low-density lipoprotein receptor (LDL-R) or LDL-R-related protein 1 (LRP1) receptor) pathways [17]. Once it reaches the lysosomes, the 52 kDa human pro-cathepsin D is proteolytically processed to the 48 kDa intermediate form that is further processed into the mature two-chain enzymatic form (a heavy 34 kDa chain and a light 14 kDa chain) [18]. This final maturation step occurs via cathepsins B and L activity [19] (Figure 1a). There is evidence that progranulin promotes in vitro maturation of pro-cathepsin D in a concentration-dependent fashion [20]. It was proposed that progranulin binds to the propeptide, destabilizing its interaction with the catalytic center of cathepsin D and promoting its autocatalytic activation cleavage [20]. Multi-granulin domain peptides (progranulin cleaved into smaller domains) showed an even more significant effect on the cathepsin D maturation, in a concentration- and pH-dependent manner [21]. The mature cathepsin D consists of two domains flanking the deep active site cleft [22]. Each domain provides a catalityc Asp to the catalytic site (Figure 1b). The two Asp active site residues are prone to deprotonation, suggesting that cathepsin D is predominantly active at pH below 5, as found in lysosomes. Nevertheless, recent data indicate that cathepsin D can also be active at higher pH in extracellular space and the cytoplasm [23,24,25].

Cathepsin D has endopeptidase activity and is responsible for the degradation of misfolded, long-lived and denatured proteins, such as the acid-denatured cathepsin L [26,27]. Moreover, it modulates the activity of diverse polypeptides, enzymes, and growth factors, and is thereby an essential regulator of cell signaling [28], while cathepsin D imbalance might play an important role in acute kidney injury [29], Huntington’s disease [30], Parkinson’s disease [31], Alzheimer’s disease [32], pancreatitis [33] and coronary events [34,35], making cathepsin D a vital protease for maintaining cellular homeostasis.

## 2. Vital Functions of Cathepsin D

Cathepsin D is ubiquitously expressed in various tissues where it is involved in protein turnover. Higher levels of cathepsin D expression in the brain, including the cortex, hippocampus, striatum, and dopaminergic neurons of the substantia nigra also suggest its essential role in the proteolysis of many altered neuronal proteins [37,38]. Long-lived post-mitotic cells, like neurons, depend on cellular homeostasis, which they preserve by an efficient proteostasis system regulating protein synthesis, folding and degradation balance [31]. Cellular health depends on the removal of damaged biological molecules, as well as on the turnover of organelles with a short lifespan. These phenomena can occur via autophagosome generation where endoplasmic reticulum membranes cloak and sequester the organelles and other intracellular materials to remove [39]. A process known as macroautophagy begins when the primary lysosome is merged with the autophagosome. In this scenario, cathepsin D deficiency has been identified as one of the causes of severe autophagy blockage in mice [40].

Cathepsin D is also necessary for proteostasis recovery after oxygen deprivation [41]. An increased protein aggregation characterizes the trophoblasts derived from patients with preeclampsia. Hypoxia-induced endoplasmic reticulum stress, one of the reasons for preeclampsia, can negatively affect transcription factor EB (TFEB) expression and its nuclear translocation leading to the decreased expression of TFEB-regulated lysosomal proteins, like lysosomal-associated membrane protein 1 (LAMP1), LAMP2, and cathepsin D. Impaired lysosomal biogenesis and autophagy resulted in increased protein aggregate accumulation in the placenta, due to the lack of lysosomal components [41]. In neurons, a low lysosomal enzymatic turnover, including cathepsin D, affects cell recovery following ischemic stroke even after mTOR activation induced by oxygen-glucose deprivation/reoxygenation [42]. In contrast, in a murine model, neuronal cathepsin D expression was shown to protect the brain against stroke injury by improving the lysosomal function [43]. Cathepsin D and cathepsin B double-knockout in mice was shown to cause impaired autophagy in the pancreas, which induced chronic pancreatitis [44]. Cathepsin D also takes part in autophagy during atherosclerosis [45].

In vitro experiments demonstrated a cathepsin D increase during apoptosis and other mechanisms of regulated cell death [46]. In this scenario, the bis-aryl urea derivative N69B demonstrated its anticancer activity by increasing cathepsin D-mediated tumor cell apoptosis through the B-cell lymphoma 2 (Bcl-2) homology domain 3 interacting domain-death agonist (Bid)/Bcl-2-like protein 4 (Bax)/cytochrome C/caspase 9/caspase 3 pathway activation [47]. Another study performed on the human neuroblastoma SH-SY5Y cell line demonstrated the cathepsin D involvement in the mitophagy process [48]. Namely, it was reported that mitophagy induction leads to enhanced expression of TFEB, which in turn, increases the synthesis of lysosomal proteins, including cathepsin D.

Cathepsin D is also involved in placenta antibacterial and antiviral defense when expressed with Napsin A [49]. They process the hemoglobin subunit β (HBB) to generate the bioactive peptide HBB_112–147_. Moreover, cathepsin D takes part in epidermal differentiation and barrier repair [50], damaged mitochondria degradation [51], promotes the fibrogenic potential in hepatic stellate cells [52] and favors cathepsin B activation [33]. It was suggested that cathepsin D might play an essential role in central nervous system (CNS) myelination since it is involved in cholesterol and lipid metabolism [53,54]. Recently, it was shown that zymogen form of cathepsin D possesses phosphatase activity in the cytosolic compartment [55]. The protease dephosphorylates cofilin, which in turn modulates actin remodelling and cell mitosis.

Despite the multiple roles of cathepsin D, the involvement of this protease in CNS proteostasis is the most investigated area. The importance of cathepsin D for the normal functioning of CNS is emphasized by many neurodegenerative diseases caused by cathepsin D deprivation. Cathepsin D deficiency resulted in fatal neurological disorders characterized by a significant loss of neurons and myelin in human infants and sheep [56], as well as in cell death in several tissues, including the brain [57], intestine, lymphoid organs [58] and the retina [59].

## 3. Cathepsin D Down-Regulation in Neurodegenerative Diseases

Reduced proteolytic activity of lysosomal cathepsins (including cathepsin D) was shown to induce a progressive accumulation of undigested autophagic substrates and unfolded or oxidized protein aggregates within the lysosomes during aging. This phenomenon can cause lysosomal membrane damage, culminating in cathepsin leakage and consequent apoptosis, favoring neurodegeneration [60].

Most neurodegenerative conditions can be distinguished in idiopathic (derived from unknown causes) and familial forms (linked to the gene mutation) [61]. Familial forms are characterized by continuous aggregation of short or immature proteins. Many mutated neuronal proteins such as amyloid precursor, α-synuclein, and huntingtin are cathepsin D substrates [32,60]. Some experiments demonstrated the cathepsin D involvement in the metabolism of cholesterol and glycosaminoglycans determining neuronal plasticity. Mutations in the *CTSD* gene are associated with different neurological defects [62]. *CTSD* gene homozygous inactivation was reported to cause human congenital neuronal ceroid lipofuscinosis (NCL) with postnatal respiratory insufficiency, epilepsy, and death within hours to weeks after birth, due to neurological defects and absence of neuronal α-synuclein accumulation [56]. Significant loss of cathepsin D enzymatic function due to the *CTSD* gene heterozygous missense mutations is associated with childhood motor and visual disturbances, cerebral and cerebellar atrophy, as well as progressive psychomotor disability. NCL3, a group of rare recessive lysosomal storage diseases, is associated with mutations in 14 different genes, including *CTSD* [63]. The main NCL3 symptoms are early blindness and severe progressive neurodegeneration. The missense variant in ceroid lipofuscinosis 5 (*CLN5*) c.A959G (p.Asn320Ser) correlates with Alzheimer’s disease occurrence [64]. More precisely, the *CLN5* c.A959G variant causes a defect in the transport of cathepsin D to the endolysosomal compartment, resulting in an increased pro-cathepsin D level and a reduced mature cathepsin D expression [64]. Mutations and polymorphisms of cathepsin D linked with the NCL are summarized in Table 1.

In line with these data, any misbalance in protein degradation processes, such as autophagy and lysosomal hydrolytic enzyme activity is linked to some age-associated neurodegenerative disorders (e.g., Parkinson’s, Alzheimer’s, Huntington’s, and Niemann-Pick disease type C (NPC)) [31]. NPC is portrayed with intracellular accumulation of lipids, among which there are mainly cholesterol and glycosphingolipids. One of the tissues affected by this disease is the brain tissue [68]. Vomeronasal neuroepithelium in an NPC1 mouse model showed virtually absence of cathepsin D reactivity [69].

Some forms of cathepsin D deficiency could also predispose to late-onset Alzheimer’s disease and Parkinson’s disease [32]. The polymorphism c.C224T (p.Ala58Val) located in exon 2 of the *CTSD* gene was associated with sporadic late-onset Alzheimer’s disease in the adult German, Iranian, and Ecuadorian populations [70]. However, in other adult populations, e.g., Spanish, Italian, Korean, and North American, this association could not be established [70]. Alzheimer’s disease histopathology is characterized by accumulations of hyperphosphorylated τ protein and amyloid β (Aβ). Mouse models of Alzheimer’s disease showed that Aβ aggregates increase astrocyte lysosome pH, resulting in a decreased cathepsin D activity and lysosome damage [71]. In support of this evidence, extralysosomal cathepsin D was detected via confocal microscopy after long-term exposure of microglia to Aβ aggregates following lysosomal membrane permeabilization [72]. Another mechanism was proposed by Suire et al. where Aβ42 inhibits cathepsin D in a low-nanomolar range, and thus, prevents cathepsin D degradation of τ protein [73]. Statistical analysis showed that cathepsin D level in plasma of patients with Alzheimer’s disease is decreased comparing to the subjects without cognitive impairment [74]. Therefore, cathepsin D can be used as a diagnostic biomarker for Alzheimer’s disease.

The unique role of cathepsin D in α-synuclein proteolysis, which accumulates in Parkinson’s disease, has been proven. Sulfated glycosaminoglycans, which hoard in substantia nigra of Parkinson’s disease patients along with α-synuclein, decrease the activity of cathepsin D [75]. The deficiency of glycosaminoglycans results in higher levels of proteolysis mediated by cathepsin D and consequent drop of α-synuclein aggregates [75]. Earlier studies demonstrated that in vitro cathepsin D yields incomplete proteolysis of α-synuclein and generates truncated C-terminal peptides, and in the acidic lysosomal lumen, enhance amyloid formation [76]. Later on, liquid chromatography-mass spectrometry analysis showed that anionic phospholipids are crucial for cathepsin D cleavage throughout α-synuclein sequence [77]. Aufschnaiter et al. showed that cathepsin D proteolysis of α-synuclein also needs calcineurin basal level expression [78].

Mutations in the β-Glucocerebrosidase gene (*GBA1*) are associated with Parkinson’s and Gaucher’s disease. Experiments on dopaminergic neurons and astrocytes carrying this mutation indicated excessive levels of α-synuclein released from neurons that are eventually endocytosed by astrocytes [79]. α-Synuclein accumulated in the lysosomes where it aggregated due to reduced cathepsin D activity. Other experiments also established decreased cathepsin D activity in substantia nigra and frontal cortex of patients with Parkinson’s disease and Lewy body dementia [80]. Conversely, some studies showed increased levels of cathepsin D in Parkinson’s fibroblasts [81] and dopaminergic neurons [82]. In this context, Puska et al. showed that cathepsin D levels are increased while α-synuclein forms pre-aggregates [83]. However, the formation of Lewy bodies decreases the cathepsin D level. A possible explanation for cathepsin D low levels in patients with Parkinson’s disease might be the down-regulation of the M6P receptor due to its compromised retrograde transport from endosomes to the trans-Golgi network [84]. The study conducted in an *ATP13A2* deficient zebrafish (Parkinson’s disease model) confirmed the reduced cathepsin D expression and showed lysosomal abnormalities, leading to the degeneration of dopaminergic neurons arguably caused by intracellular trafficking impairment [85]. Decreased levels of cathepsin D were also detected in plasma of patients with Parkinson’s disease comparing to the patients with essential tremor [86]. Therefore, all of the above data qualify cathepsin D as one of the biomarkers of Parkinson’s disease [87,88].

The possible role of cathepsin D in the development of prion diseases was demonstrated in interferon-α/β receptor knock-out mice showing decreased levels of disease-associated microglial cathepsin D and CD68 receptor, especially in white matter, which resulted in the slow disease progression [89].

Therefore, the deficiency of cathepsin D triggered by misregulation of its transport, maturation, and enzymatic activity results in the anomalous deposit of undigested cellular material in lysosomes [60]. Accumulations in lysosomes during aging weaken the lysosomal membrane causing enzymatic leakage, cell death and neurodegenerative disorders [90]. Everything aforesaid leads to the conviction that neuronal cellular health largely relies on cathepsin D-mediated proteolysis [60]. Therefore, the perspective approach in preventing the onset of a neurodegenerative disorder is to maintain a sufficient level of cathepsin D within the lysosomes.

## 4. Treatment to Restore Cathepsin D

Cathepsin D deficiency caused by mutations in the *CTSD* gene can be restored via recombinant protease as proposed by Marques et al. for NCL replacement therapy [40]. Specifically, the protocol was based on recombinant human pro-cathepsin D synthesized and purified from the human HEK 293 EBNA kidney cell line and delivered to the lysosomes, where it could be processed to its active form digesting protein aggregates. Possible difficulties to this approach were represented by non-functional M6P and LRP1 receptors and the cysteine protease activity [40].

In the case of a high level of pro-cathepsin D and a low level of the active form of the enzyme, cathepsin D maturation can be induced. As aforementioned, there is evidence that progranulin promotes in vitro maturation of pro-cathepsin D in a concentration-related process [20]. The maturation of pro-cathepsin D to its active form is stimulated even more significantly in the presence of multi-granulin domain peptides BAC and CDE resulting in an 80% active cathepsin D [21]. This might explain the reason why the progranulin gene therapy improves lysosomal dysfunction and microglial pathology associated with frontotemporal dementia and NCL [91].

Some neurodegenerative diseases are associated with a low cathepsin D level expression or activity inhibition. Eight lysosomotropic drugs (chloroquine, fluoxetine, imipramine, latrepirdine, tamoxifen, chlorpromazine, amitriptyline, and verapamil) were shown to increase cathepsin D activity at multiple concentrations after 24-h exposure [92]. A higher cathepsin D level was recorded within 4 h of latrepirdine and chlorpromazine treatment. A proteomic study in an in vivo model of depression showed that fluoxetine administration strongly up-regulated the expression of cathepsin D, proteins engaged in the improvement of learning and memory processes (stathmin 1 and dynamin-1), and proteins involved in mitochondrial biogenesis and defense against oxidative stress (protein deglycase DJ-1) [93]. *Mycobacterium tuberculosis* can inhibit phagosome maturation in infected macrophages by reducing galectin-3 expression [94]. This phenomenon can affect the development of the active cathepsin D. However, galectin-3 and cathepsin D expression could be restored by treatment with gallium encapsulated in polymeric nanoparticles favoring infection inhibition [94]. Experiments on a mouse model of Alzheimer’s disease produced evidence that cilostazol can reinstate low pH in astrocyte lysosomes, consequently suspending the inhibitory effect of Aβ on the activity of cathepsin D [71]. Chlorogenic acid is another compound proven to have a neuroprotective role in an Alzheimer’s disease mouse model [95]. Specifically, chlorogenic acid was shown to up-regulate cathepsin D and other cathepsins expression via the mTOR/TFEB signaling pathway in APP/PS1 mice (Alzheimer’s disease model) and Aβ_25-35_-exposed SH-SY5Y cells [95]. It was shown that glucocerebrosidase replacement or chaperone therapy of *GBA1*-mutant resulted in restored cathepsin D protein levels and activity, leading to decreased levels of monomeric α-synuclein in *GBA1-*mutant neurons [96].

There are multiple approaches to increase active cathepsin D in the cells. However, it is necessary to understand the kind of mechanism that caused the protease deprivation at first. There are also some risks associated with the modulation of this fragile balance that could favor the onset of other disorders.

## 5. Excessive Levels of Cathepsin D in Neurodegenerative Disorders

As mentioned earlier, Aβ aggregates can decrease the cathepsin D activity within the lysosomes. However, high exosomal levels of cathepsin D and LAMP1 together with low levels of the 70 kDa heat shock proteins were detected in the blood of patients with Alzheimer’s disease, suggesting a diagnostic role for this protease [97]. In addition, it was shown that the levels of cathepsin D are similar among patients with mild and severe Alzheimer’s disease and mild cognitive impairment [98]. On the other hand, the distinction between these groups of patients can be determined by comparing cathepsin B and cathepsin S levels [98]. Although, these results are inconsistent with the above-mentioned ones, so further research is required [74].

Glutaric acidemia type I (GA1) is a chronic progressive neurodegeneration caused by severe deficiency of glutaryl-CoA dehydrogenase activity, leading to glutaric acid and glutarylcarnitine accumulation [99]. It was demonstrated that brain-derived neurotrophic factor and cathepsin D significantly increased in the plasma of GA1 patients compared to the control group. Also, a positive correlation was found between the levels of cathepsin D and glutarylcarnitine levels that reflected the accumulation of glutaric acid. These data support the theory that glutaric acid is a critical player in the occurrence of neurological damage in GA1 patients [99].

The plasma of maple syrup urine disease patients showed increased levels of cathepsin D compared to the control group [100]. The authors proposed that high levels of cathepsin D may result from its role in cytokines- and oxidative stress-induced apoptosis. Increased cathepsin D levels were also observed in neurofibrillary tangles of parietal cortex neurons, where it correlated with hyperphosphorylated τ protein, but did not co-localize with α-synuclein inclusions [101].

Most neurodegenerative disorders caused by cathepsin D imbalance are characterized by down-regulation of the protease. However, there is evidence that in pathological conditions it can increase extracellularly, while the lysosomal concentration of cathepsin D can decrease.

## 6. Excessive Levels of Cathepsin D in Disorders Associated with Diabetes

Hyperglycemia can trigger cathepsin D release from the lysosomes by inducing lysosomal membrane permeabilization and ion release. Cathepsin D can remain active in non-acidic pH environments such as the cytosol. Altogether, this contributes to hyperglycemia-induced cardiomyocyte injury in patients with diabetic cardiomyopathy [102]. These results correlate with Hoef et al.’s study, which concluded that higher circulating cathepsin D levels correlate with greater heart failure severity [103]. Also, Liu et al. demonstrated a correlation between increased cathepsin D levels and type 2 diabetes [104].

The data concerning the influence of excessive cathepsin D activity outside the lysosomes on the severity of diabetes are relatively scarce. Further investigation of the mechanisms of cathepsin D regulation in this disorder may reveal new approaches for anti-diabetic therapy. Nevertheless, accumulated data detail the connection between lysosomal protease release and a pathological condition.

## 7. Excessive Levels of Cathepsin D in Malignant Tumors

Increased extracellular levels of cathepsins are well-characterized for tumors, and cathepsin D is no exception. High levels of cathepsin D were observed in breast [105], ovarian [106], colorectal [51], prostate, bladder cancer [107] and melanoma [108]. Numerous studies found that cathepsin D level may represent an independent prognostic factor in many cancers and is considered a potential target of anticancer therapy [109]. Cathepsin D was shown to have pro-angiogenesis, pro-apoptotic, pro-invasive and pro-metastatic properties [28]. Some research suggests the roles of cathepsin D in cancer cells in maintaining lysosomal integrity, redox balance and nuclear factor erythroid 2-related factor 2 activity, thus, promoting tumorigenesis [110]. There is evidence that even mutated cathepsin D deprived of catalytic activity can still have mitogenic properties by activating an unknown cell surface receptor [111].

Progesterone receptor isoforms A and B (PR-A and PR-B) ratio is used as a prognostic factor in breast cancer [112]. Breast cancer cells expressing PR-A show increased levels of proteins involved in the citric acid cycle, glycolysis, the Rho family of guanosine triphosphatase signaling, and ribonucleic acid (RNA) metabolism. Pateetin et al. showed an increased cathepsin D level in cells expressing progesterone-liganded PR-A [113]. Cathepsin D is secreted in estrogen-dependent and estrogen-independent types of breast cancer [114,115], and for this reason, represents a significant prognostic marker. Estradiol-mediated enhanced secretion of pro-cathepsin D in breast cancer cells was also established [116]. The suggested mechanism supports the involvement of the cation-dependent M6P receptor, which ensures the proper localization of the enzyme to lysosomes in MCF-7 cells (breast cancer) [116].

Liu et al. showed that tetrabromobisphenol A (TBBPA) could increase the extracellular and decrease the intracellular levels of cathepsins D and B in hepatocellular carcinoma (HCC) cell line, HepG2 [117]. These results imply that TBBPA might promote lysosomal exocytosis and consequent in vitro HepG2 cell invasion and metastasis. It is believed that TBBPA could bind mucolipin-1, forming a complex which significantly increases Ca^2+^-mediated lysosomal exocytosis in HCC [117]. Increased lysosomal membrane permeabilization in HCC could result from suppressed sulfatase 2, which leads to lysosome-associated protein transmembrane 4β inhibition whose expression depends on sulfatase 2 [118]. Detection of extralysosomal cathepsin D represents a sign of disrupted autophagy.

The outcome of several studies claims that cathepsin D, together with cathepsin B, plays essential roles in the production of angiotensin peptides in glioblastoma cells bypassing the renin-angiotensin system [119]. Basu et al. demonstrated that colorectal cancer cells with overexpression of immunoglobulin-like cell adhesion receptor L1 also showed increased extra- and intra-cellular levels of cathepsin D [120]. They suggested the essential role of cathepsin D in colorectal cancer progression. Cathepsin D might represent a therapeutic target for curing invasive colorectal cancer, given that it is only detected in invasive areas of the tumor [120].

The abundance of cathepsin D outside the lysosomes in malignant tumors makes it a convenient marker and a target for cancer treatment. Still, there must be targeted delivery of the inhibitors to prevent the onset of off-site disorders.

## 8. Cathepsin D Inhibitors

Most cathepsin are cysteine proteases so their inhibitors belong to the cystatin superfamily, including stefins, kininogens, thyropins, and serpins [5]. However, there are no known endogenous inhibitors for the ascpartic protease cathepsin D in mammals [121]. Nonetheless, there are several natural inhibitors isolated from other species. The most known inhibitor of cathepsin D is pepstatin A from Actinomyces, an inhibitor of aspartic proteases [76]. Baldwin et al. consucted the thorough comparative analysis of the native protease and the protease in complex with pepstatin A (PDB ID: 1LYA, 1LYB) [122]. However, non-specific inhibition of aspartic proteases, including cathepsin D, may induce side effects (e.g., CNS), and may not be safe as therapeutics [123]. Therefore, novel studies examine new possible inhibitors using a 2/3-dimensional quantitative structure-activity relationship, representing a powerful tool for explaining the relationships between chemical structure and experimental observations [124]. These studies showed that oxymatrine from *Sophora flavescens* down-regulated the expression of cathepsin D, inhibiting high mobility group protein 1/toll-like receptor 4/nuclear factor κ-light-chain-enhancer of activated B cells (NF-κB) signaling pathway, resulting in the suppression of microglia-mediated neuroinflammation in a 1-methyl-4-phenyl-1,2,3,6-tetrahydropyridine (MPTP)-stimulated mouse model of Parkinson’s disease [125]. Fucoidan from brown algae inhibited the expression of cathepsin D and Bax in the murine dopaminergic nerve precursor cell line, MN9D (neuroblastoma), and indicated its prospective role as a neuroprotective agent in Parkinson’s disease treatment [126]. Flavonoid morin hydrate from *Maclura pomifera*, *Maclura tinctoria*, and *Psidium guajava* is claimed to be anti-oxidative and anti-inflammatory. Recently, it was shown to inhibit cathepsin D in kidneys of mice with chronic kidney damage [127]. Molecular docking revealed that morin hydrate interacts with cathepsin D active site via H-bonds and hydrophobic interactions.

Some experiments demonstrated that a small-molecule inhibitor of β-secretase 1, aminothiazole-based compound LY2811376, inhibits cathepsin D activity as well [128]. This pH-dependent suppression results from a salt bridge formation between the inhibitor aspartyl binding motif and Asp_33_ in cathepsin D. Furthermore, a group of scientists using a graph convolutional neural network (CNN) indicated that cathepsin D is an off-target ligand of some β-secretase 1 inhibitors [129]. Another work unraveled the neuroprotective role of necrostatin-1 against oxidative stress-induced cell damage [130]. It demonstrated that necrostatin-1 induced cathepsin D inhibition in a cell line of differentiated human neuroblastoma RA-SH-SY5Y. Recently, pseudo-dipeptide binding motif of pepstatin A was used to design macrocyclic peptidomimetic inhibitors [36]. They were shown to inhibit cathepsin D in nanomolar and sub-nanomolar range without citotoxicity in contrast to pepstatin A (PDB ID: 6QBG, 6QBH, 6QCB). The high throughput screening (HTS) identified a series of acylguanidine inhibitors which interact with catalytic Asp via H-bonds and inhibit cathepsin D with nanomolar potency [131]. However, they were characterized as inhibitors with low microsomal stability and permeability so further optimization is required (PDB ID: 4OBZ, 4OC6, 4OD9). Further optimization of the compounds resulted in the inhibitor 24e with improved microsomal stability, as shown on human and mouse liver microsomes [132].

Several studies indicated the effectiveness of combination therapy with natural and synthetic inhibitors. For instance, co-treatment with praeruptorin C from *Peucedanum praeruptorum* and U0126 synergistically inhibited cathepsin D expression through the extracellular signal-regulated kinase 1/2 signaling pathway in human non-small cell lung cancer cells [133]. There is evidence that autophagy modulators chloroquine, KU-55933, and rapamycin from *Streptomyces hygroscopicus* combined with a recombinant analog of human milk protein lactaptin decreased cathepsin D activity with cytotoxic effects in MDA-MB-231 cell line (breast carcinoma) [134].

RNA-based compounds might represent another approach to inhibit cathepsin D. It was shown that cathepsin D knockdown via *CTSD* shRNA lentiviral vector transduction suppressed lipopolysaccharide-induced neuroinflammation by inhibiting NF-κB signaling pathway [135]. This phenomenon was obtained by regulating the nuclear factor NF-κB p65 subunit (p65) nuclear translocation both in MPTP-challenged mice and lipopolysaccharide-induced murine microglia, BV-2 cell line [135]. Phosphatidylinositol 3-phosphate (PtdIns(3)*P*) RNA aptamer binding to PtdIns(3)*P* inhibited autophagy by hampering lysosomal acidification, which resulted in reduced cathepsin D activity [136].

Although some studies have shown that cathepsin inhibitors may be used as metastasis suppressors [137], others suggested that suppression of cystatins may contribute to the suppression of breast cancer [138], colorectal cancer [139] pancreatic ductal adenocarcinoma [140]. Therefore, one has to be careful in developing a treatment, in order to avoid side effects. Nevertheless, this should not be an issue since cathepsin D inhibitors are represented by synthetic and natural compounds including peptides and RNAs, providing an extensive portfolio of therapies depending on the mechanisms of up-regulation, drug delivery, as well as severity of the disease. Cathepsin D inhibitors are summarized in Table 2.

## 9. Conclusions

Cathepsin D is involved in autophagy, endocytosis, degradation of misfolded or mutated proteins, regulation of the activity of various polypeptides, enzymes and growth factors. Due to its numerous roles in metabolic processes, necessary for cell survival and death, it is crucial to maintain the level of cathepsin D activity within optimal limits.

Decreased cathepsin D activity and/or levels have been observed in several neurodegenerative disorders (Figure 2). Replacement therapy with recombinant pro-cathepsin D seems a perspective approach to mitigate its reduced expression [40]. When it comes to compensating cathepsin D reduced activity due to lysosome membrane permeability increase, the therapy should aim to maintain membrane homeostasis, preserving the pH and cholesterol levels in these organelles [151]. It is also necessary to explore the signaling pathways that lead to a change in lysosome membrane proteins conformation to develop appropriate inhibitors.

Lysosome membrane permeabilization also leads to excessive cathepsin D activity in the cytoplasm and extracellular space. This contributes to malignant tumor growth, metastasis and angiogenesis (Figure 2). The inhibitors may decrease cathepsin D activity, but the research for new, specific inhibitors should continue. It is necessary to emphasize that one should be extremely careful in using inhibitors for therapeutic purposes. As previously mentioned, cathepsin D participates in many processes in the whole organism. Therefore, the systematic use of inhibitors should be avoided, and targeted therapy should be investigated hand in hand with the development of novel therapeutics [152].

There are still several functions of cathepsin D to investigate. For example, cathepsin D is involved in the development of disorders associated with diabetes or its involvement in redox response suggesting the potential use of this protease in anti-photoaging therapies [50]. Therefore, there is an abundance of pathways for exploring future research in consideration of the many and various roles of cathepsin D.

## Figures and Tables

**Figure 1 pharmaceutics-13-00837-f001:**
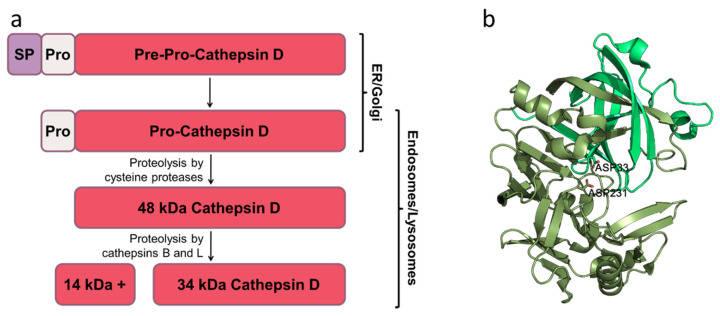
Cathepsin D maturation and structure: (**a**) Major steps of cathepsin D maturation. The description is in the text. SP—signal peptide; Pro—prodomain; kDa—kilodaltons; ER—endoplasmic reticulum. (**b**). Mature cathepsin D structure. The two domains are indicated in different shades of green. Catalytic Asp are represented as sticks. The structure was obtained from Protein Data Bank (PDB ID: 6QCB) [36]. The figure was made in the PyMOL Molecular Graphics System, Version 1.2r3pre, Schrödinger, LLC.

**Figure 2 pharmaceutics-13-00837-f002:**
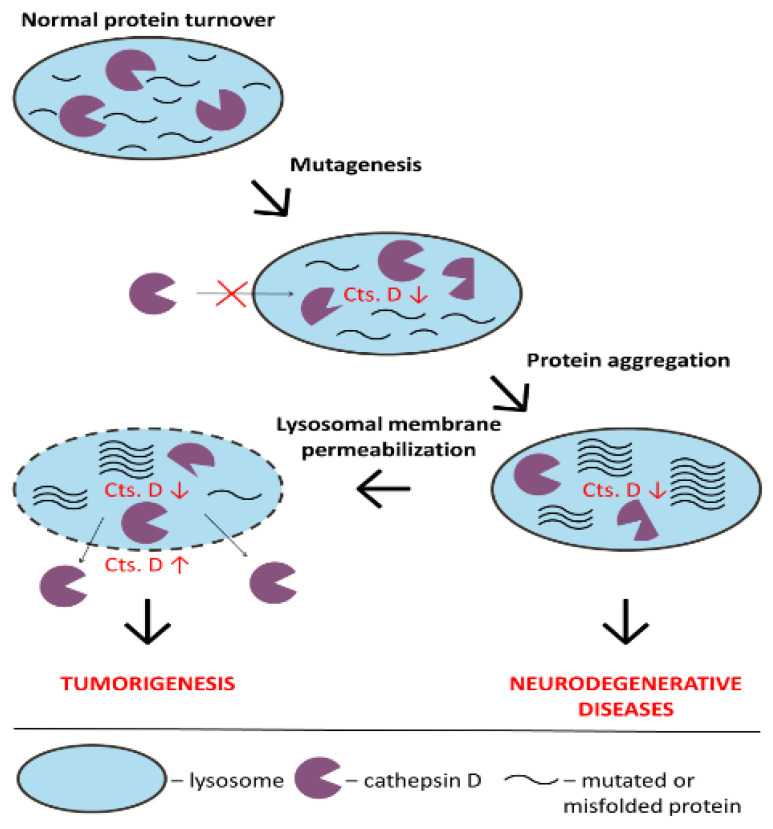
Role of Cathepsin D in cancer and neurological disorders. Cathepsin D mutations may cause a decrease in its lysosomal traffic and activity, leading to protein aggregation in the organelles and neurodegenerative diseases development. Protein aggregation may cause lysosomal membrane permeabilization with the subsequent increase of cathepsin D outside the lysosomes. On the other hand, the excessive extracellular activity of cathepsin D can contribute to tumorigenesis.

**Table 1 pharmaceutics-13-00837-t001:** Cathepsin D mutations related to neuronal ceroid lipofuscinosis (NCL).

DNA ^1^ /Protein Change	Type of Mutation	Effect on Protease	NCL Type	Ref.
c.392A>G/p.Tyr131Cys	Missense	Reduced enzymatic activity	Late infantile NCL (LINCL)	[65]
c.446G>T/p.Gly149Val	Missense	Reduced enzymatic activity	Juvenile NCL (JNCL)	[66]
c.1196G>A/p.Arg399His				
c.299C>T/p.Ser100Phe	Missense	Reduced enzymatic activity	Congenital CLN ^2^ (CLN10)	[67]
c.764dupA/p.Tyr255	Nonsense	Absence of protein	CLN10	[56]
c.6517T>A/p.Phe229Ile	Missense	Reduced protein amount and enzymatic activity	NCL-like disorder	[63]
c.10267G>C/p.Trp383Cys				

^1^ deoxyribonucleic acid; ^2^ ceroid-lipofuscinosis, neuronal.

**Table 2 pharmaceutics-13-00837-t002:** Cathepsin D inhibitors in chronological order.

Inhibitor	Mechanism	Ref.
**Natural Compounds**		
Pepstatin A from *Actinomycetes*	Non-competitive inhibitor	[141]
Cycloheximide from *Streptomyces griseus*	Protein synthesis inhibitor	[142]
The 22-kDa cathepsin D inhibitor protein of potatoes (PDI) from *Solanum tuberosum*	Reversible inhibitor	[143]
Equistatin from *Actinia equina*	Reversible inhibitor	[144]
Fucoidan from brown seaweeds and algae	Down-regulator of the expression	[126]
Oxymatrine from *Sophora flavescens*	Down-regulator of the expression	[125]
Morin hydrate from *Maclura pomifera*, *Maclura tinctoria*, *Psidium guajava*	Reversible inhibitor	[127]
**Synthetic compounds**		
Dithiophosgene	Irreversible covalent inhibitor	[145]
2,2-Dichloro-1,3-dithiacyclobutanone		
Diazo compounds	Irreversible covalent inhibitor	[146]
Pro-Pro-Phe-Phe-Val-D-Leu	Reversible inhibitor	[147]
Cbz-Val-Val-(3S4S)-statine	Reversible inhibitor	[148]
Ibu-His-Pro-Phe-HCys-Sta-Leu-NH-[CH_2_]_2_-S-Acm	Reversible inhibitor	[149]
Derivatives of 4-(morpholinylsulphonyl)-L-Phe-P2-(cyclohexyl)Ala psi[isostere]-P1′-P2′	Irreversible covalent inhibitor	[150]
Lentiviral shRNA constructs	RNA interference inhibitor	[76]
Acylguanidines	Reversible inhibitor	[131,132]
LY2811376	Reversible inhibitor	[128]
PtdIns(3)*P* ^1^ RNA aptamer	Inhibitor of PtdIns(3)*P*	[136]
Macrocyclic inhibitors	Competitive inhibitor	[36]
Necrostatin-1	Suppressor of activity	[130]
**Polytherapy**		
RL2 ^2^, with chloroquine, Ku55933, and rapamycin from *Streptomyces hygroscopicus*	Suppressor of activity	[134]
Praeruptorin C from *Peucedanum praeruptorum* and U0126	Inhibitor through ERK1/2 ^3^ signaling pathway	[133]

^1^ phospatidylinositol 3-phosphate; ^2^ recombinant analog of human milk protein lactaptin; ^3^ extracellular signal-regulated kinase 1/2.

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
