# Peer review of "Cathepsin D—Managing the Delicate Balance"

_pharmaceutics, 2021, doi:10.3390/pharmaceutics13060837_

Round 1

Reviewer 1 Report

This manuscript is a concise summary of up-to-date information of cathepsin D putting some weight on the pathological impact.

Overall, this is informative providing certain merits to the research area especially in the neurodegeneration. The contents were nicely organized and intriguing. However, multiple misuse or incorrect citations are found through entire manuscript. The citation style is perplexing and bothers readers. Some citations could be inserted to a little earlier at the first appearance of each statement. 

I would ask the authors to scrutinize all references once again and reconsider where to insert the citations.

Below are some specific points to consider (including, but not limited to):
Line 108-113 only if this statement is also derived from the reference 39
Line 125-126 reference needed
Line 131-133 the citations could be more definitive to each statement.
Line 139-142 the references #51 and 52 seem to be incorrect. 
Line 164-166 the reference #57 could be cited for this statement.
Line 187-189 incorrect statement: the experiment in the ref #65 suggested the extralysosomal cathepsin-D but extracellular localization.
Line 210-211 “But the formation of Lewy bodies decreases the cathepsin D level, and activity decreased [74].” Is this statement true? The referenced article shows the alteration of immunoreactivity, but I found no direct evidence showing the reduction of cathepsin D enzymatic activity.
Line 232-238 The Marques’ reference [38] should be inserted at the first sentence (Line 233).
Line 267 What does GCase stand for?
Line 278-280 reference needed
Line 291-292 reference #90 should be cited here.
Line 337-339 reference #107 should be cited here.
Line 359-360 The cystatins are a family of “cysteine” protease inhibitors but not for aspartic protease. This introduction could be a little misleading. Please reconsider.

Figure 2 is a little reader unfriendly. Does the big circle indicate lysosome or cell? Is the purple oval cathepsin D? Does the wavy line indicate mutated proteins? Please clarify what each item exactly represent.

Author Response

Dear Editor and Reviewers,
Thank you for your beneficial suggestions. We have fully addressed your concerns below and have thoroughly revised the manuscript based on your recommendations.

All changes have been marked in red.

This manuscript is a concise summary of up-to-date information of cathepsin D putting some weight on the pathological impact.

Overall, this is informative providing certain merits to the research area especially in the neurodegeneration. The contents were nicely organized and intriguing. However, multiple misuse or incorrect citations are found through entire manuscript. The citation style is perplexing and bothers readers. Some citations could be inserted to a little earlier at the first appearance of each statement. 

I would ask the authors to scrutinize all references once again and reconsider where to insert the citations.

Author response: We thank the reviewer for these observations. We have rigorously edited the references in line with these recommendations (lines 77, 79, 95-97, 118, 139, 143, 145-147, 157, 158, 172, 182, 200, 217, 229, 233-235, 260, 278, 284, 291, 308, 314, 321, 346, 347, 364, 369, 375, 381, 415, 420, 443, 449-451).

Below are some specific points to consider (including, but not limited to):
Line 108-113 only if this statement is also derived from the reference 39
Line 125-126 reference needed
Author response: The references have been relocated where necessary (lines 118, 139).

Line 131-133 the citations could be more definitive to each statement.
Author response: The references have been allocated to the proper statements (lines 145-147).

Line 139-142 the references #51 and 52 seem to be incorrect. 
Author response: The references have been relocated and added to support each statement in the sentence (lines 157, 158).

Line 164-166 the reference #57 could be cited for this statement.
Author response: The reference has been added for the statement (line 182).

Line 187-189 incorrect statement: the experiment in the ref #65 suggested the extralysosomal cathepsin-D but extracellular localization.
Author response: The statement has been corrected (line 206).

Line 210-211 “But the formation of Lewy bodies decreases the cathepsin D level, and activity decreased [74].” Is this statement true? The referenced article shows the alteration of immunoreactivity, but I found no direct evidence showing the reduction of cathepsin D enzymatic activity.
Author response: The statement has been corrected according to the reference article (line 236).

Line 232-238 The Marques’ reference [38] should be inserted at the first sentence (Line 233).
Line 267 What does GCase stand for?
Author response: The reference has been added in the first sentence in the paragraph (line 260). The abbreviation has been replaced with the full form of the word (line 294).

Line 278-280 reference needed
Author response: Thank you.The reference has been added (line 308).

Line 291-292 reference #90 should be cited here.
Author response: The reference has been relocated (line 321).

Line 337-339 reference #107 should be cited here.
Author response: The reference has been added (line 369).

Line 359-360 The cystatins are a family of “cysteine” protease inhibitors but not for aspartic protease. This introduction could be a little misleading. Please reconsider.

Author response: The paragraph has been edited to clarify the message (lines 389, 391, 448-452).

Figure 2 is a little reader unfriendly. Does the big circle indicate lysosome or cell? Is the purple oval cathepsin D? Does the wavy line indicate mutated proteins? Please clarify what each item exactly represent.

Author response: Figure 2 has been corrected. Each item is labeled in the revised manuscript.

Reviewer 2 Report

This is a short well written review on the Cathepsin D.

However, one of the most important thing is missing – the description of the structure. Any scientist involved in the drug design process, will seek for the review first. In this one he/she will find lots of useful informations on cathepsin D. The other step is the seek for the structural analysis, which is crucial for a drug design. There are several structures deposited in PDB database, which can be used for a detail description of the inhibition mechanism. There are other inhibitors, which are not mentioned in the manuscript with available structures. Why?

Please add one more figure with a protein structure, information on available PDB structures and some structural information on the inhibition mechanism.

Author Response

Dear Editor and Reviewers,
Thank you for your very helpful suggestions. We have fully addressed your concerns below and have thoroughly revised the manuscript based on your recommendations.

All changes have been marked in red.

This is a short well-written review on the Cathepsin D.

However, one of the most important thing is missing – the description of the structure. Any scientist involved in the drug design process, will seek for the review first. In this one he/she will find lots of useful informations on cathepsin D. The other step is the seek for the structural analysis, which is crucial for a drug design. There are several structures deposited in PDB database, which can be used for a detail description of the inhibition mechanism. There are other inhibitors, which are not mentioned in the manuscript with available structures. Why?

Please add one more figure with a protein structure, information on available PDB structures and some structural information on the inhibition mechanism.

Author response: We thank the reviewer for these observations. We have studied the structures of cathepsin D deposited in the protein database implementing Figure 1 with the structure of cathepsin D and a short description (see figure 2 and lines 83-85). We also add the information of more inhibitors (see table 2 and lines 394-396, 409-413, 422-431).

Reviewer 3 Report

This review on the role of cathepsin D is brief but well thought out and the main aspects of the function performed by cathepsin D are considered. However, a review should be updated where possible with the latest publications on the subject. I suggest inserting them in the text and references e.g. :
a- Dai J, Zhang Q, Wan C, Liu J, Zhang Q, Yu Y, Wang J. Significances of viable synergistic autophagy-associated cathepsin B and cathepsin D (CTSB / CTSD) as potential biomarkers for sudden cardiac death. BMC Cardiovasc Disord. 2021 May 8; 21 (1): 233. doi: 10.1186 / s12872-021-02040-3.
b- Bunk J, Prieto Huarcaya S, Drobny A, Dobert JP, Walther L, Rose-John S, Arnold P, Zunke F. Cathepsin D Variants Associated With Neurodegenerative Diseases Show Dysregulated Functionality and Modified α-Synuclein Degradation Properties. Front Cell Dev Biol. 2021 Feb 11; 9: 581805. doi: 10.3389 / fcell.2021.581805.
c- Suire CN, Leissring MA. Cathepsin D: A Candidate Link between Amyloid β-protein and Tauopathy in Alzheimer Disease. J Exp Neurol. 2021; 2 (1): 10-15.
d- Kim JW, Jung SY, Kim Y, Heo H, Hong CH, Seo SW, Choi SH, Son SJ, Lee S, Chang J. Identification of Cathepsin D as a Plasma Biomarker for Alzheimer's Disease. Cells. 2021 Jan 12; 10 (1): 138. doi: 10.3390 / cells10010138.
e- Goyal S, Patel KV, Nagare Y, Raykar DB, Raikar SS, Dolas A, Khurana P, Cyriac R, Sarak S, Gangar M, Agarwal AK, Kulkarni A. Identification and structure-activity relationship studies of small molecule inhibitors of the human cathepsin D. Bioorg Med Chem. 2021 Jan 1; 29: 115879. doi: 10.1016 / j.bmc.2020.115879.

Author Response

Dear Editor and Reviewers,
Thank you for your very helpful suggestions. We have fully addressed your concerns below and have thoroughly revised the manuscript based on your recommendations.

All changes have been marked in red.

This review on the role of cathepsin D is brief but well thought out and the main aspects of the function performed by cathepsin D are considered. However, a review should be updated where possible with the latest publications on the subject. I suggest inserting them in the text and references e.g. :
a- Dai J, Zhang Q, Wan C, Liu J, Zhang Q, Yu Y, Wang J. Significances of viable synergistic autophagy-associated cathepsin B and cathepsin D (CTSB / CTSD) as potential biomarkers for sudden cardiac death. BMC Cardiovasc Disord. 2021 May 8; 21 (1): 233. doi: 10.1186 / s12872-021-02040-3.
b- Bunk J, Prieto Huarcaya S, Drobny A, Dobert JP, Walther L, Rose-John S, Arnold P, Zunke F. Cathepsin D Variants Associated With Neurodegenerative Diseases Show Dysregulated Functionality and Modified α-Synuclein Degradation Properties. Front Cell Dev Biol. 2021 Feb 11; 9: 581805. doi: 10.3389 / fcell.2021.581805.
c- Suire CN, Leissring MA. Cathepsin D: A Candidate Link between Amyloid β-protein and Tauopathy in Alzheimer Disease. J Exp Neurol. 2021; 2 (1): 10-15.
d- Kim JW, Jung SY, Kim Y, Heo H, Hong CH, Seo SW, Choi SH, Son SJ, Lee S, Chang J. Identification of Cathepsin D as a Plasma Biomarker for Alzheimer's Disease. Cells. 2021 Jan 12; 10 (1): 138. doi: 10.3390 / cells10010138.
e- Goyal S, Patel KV, Nagare Y, Raykar DB, Raikar SS, Dolas A, Khurana P, Cyriac R, Sarak S, Gangar M, Agarwal AK, Kulkarni A. Identification and structure-activity relationship studies of small molecule inhibitors of the human cathepsin D. Bioorg Med Chem. 2021 Jan 1; 29: 115879. doi: 10.1016 / j.bmc.2020.115879.

Author response: We thank the reviewer for these observations.  We have included all the proposed references (lines 131, 172, 208-213, 310, 311, 429-431 and Table 2). We also added more latest publications (lines 129-131, 149-151, 242-244, 409-413 and Table 2).